# Prevalence of *Schistosoma bovis* and *Schistosoma haematobium* hybrids in endemic communities in Ghana

Yvonne Ashong[1,2], Frank Twum Aboagye[3], Isaac Owusu-Frimpong[3], Samuel Nyarko[1], Jewelna Akorli[1], Linda Batsa Debrah[2], Samuel Armoo[3], Alex Yaw Debrah[4], Dziedzom K. de Souza[1]*, Mike Yaw Osei-Atweneboana[3,5]

**1** Department of Parasitology, Noguchi Memorial Institute for Medical Research, College of Health Sciences, University of Ghana, Legon, Ghana, **2** Department of Clinical Microbiology, Kwame Nkrumah University of Science and Technology, Kumasi, Ghana, **3** Biomedical and Public Health Research Unit, Water Research Institute, Council for Scientific and Industrial Research, Accra, Ghana, **4** Department of Medical Diagnostics, Kwame Nkrumah University of Science and Technology, Kumasi, Ghana, **5** CSIR-College of Science and Technology, Accra, Ghana

* ddesouza@noguchi.ug.edu.gh

## Abstract

### Background

Schistosomiasis, a debilitating parasitic disease caused by *Schistosoma* species, poses significant public health challenges in tropical regions, particularly in sub-Saharan Africa. There is growing evidence of species hybridization which may affect transmission dynamics, host range, and treatment resistance and hamper effective control strategies. This study investigated the prevalence and genetic diversity of *Schistosoma bovis* and *Schistosoma haematobium* hybrids in endemic communities in Ghana.

### Methods

A cross-sectional study was conducted with urine samples collected from 840 schoolchildren in Ga South and Birim North districts. Microscopy was used for initial screening, followed by molecular characterization and sequencing to confirm species identification and detection of hybrids. Prevalence rates were calculated, and genetic analyses were performed using phylogenetic methods based on the COX 1 gene of *Schistosoma bovis* and *Schistosoma haematobium* hybrids.

### Results

Of 840 samples analysed, 96 tested positive for *S. haematobium*, yielding a prevalence of 11.42% [$CI_{95}$: 9.26–13.96]. The highest prevalence in Ga South was among 13–17-year-olds [23.27%; $CI_{95}$: 16.52–31.77], while in Birim North, it was among 5–9-year-olds [4.82%; $CI_{95}$: 1.96–11.75]. The highest prevalence of infection in both

**Data availability statement:** All relevant data are within the paper and its Supporting information files.

**Funding:** DFID 203089, Department for International Development, UKAID The funders had no role in study design, data collection and analysis, decision to publish, or preparation of the manuscript.

**Competing interests:** The authors have declared that no competing interests exist.

districts: 22.58% [95% CI: 17.53–28.60] in Ga South and 4.23% [95% CI: 2.26–7.83] in Birim North was reported in males. Molecular characterization revealed that 33.33% [32/96, $CI_{95}$: 24.70–43.27] of the samples initially diagnosed as *S. haematobium* were identified to be *S. bovis* and 11.49% [11/96, $CI_{95}$: 6.56–19.39] of cases were hybrids.

## Conclusion

The findings of this study highlight the prevalence of *S. bovis* and *S. haematobium* hybrids in Ghana, emphasizing the importance of accurate diagnostics and targeted control measures. Continued surveillance and research are essential to address the emerging challenges posed by schistosomiasis in endemic regions.

## 1. Background

Schistosomiasis remains one of the most significant public health challenges in tropical and subtropical regions, particularly in Sub-Saharan Africa [1,2]. Caused by trematode worms of the genus *Schistosoma*, this chronic disease has profound implications for human health and economic stability [3,4]. The World Health Organization estimates that schistosomiasis affects over 200 million people worldwide, leading to about 200,000 deaths annually and contributing significantly to the global burden of disease in terms of disability-adjusted life years (DALYs) [5,6]. The disease's impact is especially pronounced in endemic areas, where it exacerbates poverty, hinders educational attainment, and impairs economic productivity [7].

In humans, the most common schistosome species include *S. mansoni*, *S. haematobium*, *S. intercalatum*, *S. guineensis*, *S. japonicum*, and *S. mekongi* [8]. In Ghana, *S. haematobium* and *S. mansoni* are the predominant species, with *S. haematobium* responsible for urogenital schistosomiasis, leading to significant morbidity and long-term health consequences [9,10]. The disease affects individuals and poses significant challenges to public health systems, particularly in areas with inadequate water and sanitation infrastructure.

Schistosomiasis is not only a human health concern but also a veterinary issue, with various *Schistosoma* species affecting livestock [11,12]. Notably, *Schistosoma bovis*, which primarily infects cattle, is gaining recognition as an emerging health threat that can impact food security and economic stability [12,13]. The phylogenetic relationship between *S. bovis* and *S. haematobium* raises concerns about potential hybridization, as these species share overlapping geographical distributions [14]. The emergence of hybrids between *S. haematobium* and *S. bovis* has been documented in several regions, raising questions about their epidemiological significance and potential impact on disease dynamics [13,15].

The identification of hybrid forms is particularly concerning due to their potential to alter transmission patterns, expand host ranges, and exhibit increased virulence, complicating control efforts [16]. Despite the increasing awareness of hybridization in schistosomes globally, there is a notable lack of data regarding *Schistosoma* hybrids

in Ghana. Hybridization presents challenges to disease prevention programs, diagnosis, and the efficacy of control and treatment measures, as hybrid forms may exhibit enhanced virulence and resistance to treatment relative to their parent species [17,18]. Understanding the presence and prevalence of these hybrids is essential for effective disease management and control strategies. Accurate identification of *Schistosoma* species and their hybrids is vital for developing targeted interventions that consider the complexities of transmission dynamics.

This study investigates the prevalence and genetic diversity of *S. bovis*, *S. haematobium* and potential hybrids in urine samples from community members in two endemic areas in Ghana. By using molecular techniques, we aimed to fill the existing knowledge gap regarding *Schistosoma* hybridization in Ghana, contributing to an understanding of the epidemiology of schistosomiasis in the region.

## 2. Methodology

### 2.1 Study area

The study was conducted in the Ga South Municipality (Tomefa, Galilea, and Domeabra communities) in the Greater Accra Region and Birim North District (Mamanso, Abirem, and Ntronang communities) in the Eastern Region. The Ga South Municipality is situated along the Weija Lake formed because of the damming of the Densu River to collect water for treatment and supply to major parts of the Greater Accra Region. Communities within the Municipality are endemic for both *S. haematobium* and *S. mansoni* due to their reliance on the lake water for their daily activities [19,20]. The Birim North District in the Eastern Region is drained by several rivers and streams and among them are the two great historical rivers, the Birim and Pra. The tributaries of these rivers in the district include the Nwi, Suten, Mamang, Adechensu, Sukrang, Nkwasua, Nyanoma, Afosu, etc. Access to potable water is a major challenge, except for New Abirem which has pipe-borne water as source of drinking water and some few communities that have boreholes. Majority of the population therefore depend mainly on wells, streams and rivers for their source of water for drinking and household activities.

### 2.2 Study population

School children (5–17 years old) were selected from schools in the two districts. Cochran's sample size formula was used in calculating for the total participants to be enrolled into the study using a schistosomiasis prevalence of 44.2% reported in Ga South district [20]. The sample size for the study was calculated as shown below per district;

$$n = \frac{Z^2 P(1-P)}{d^2}$$

where: Z is standard normal variate at 5% type 1 error (p < 0.05) which is 1.96; P is expected proportion in population based on previous studies or pilot studies; d is absolute error or precision of 5% if the prevalence of the disease is going to be between 10% and 90% [21].

$$n = \frac{Z^2 P(1-P)}{d^2} \implies n = \frac{1.96^2 \; x \; 0.442(1-0.442)}{0.05^2} \implies n = 378.99$$

The calculated sample size (n) was 379 but to ensure that the required level of accuracy was provided in the occurrence of no-response to provide sample for analysis or participant redrawing from the study, a non–response of 7% (r = 0.05) was adopted:

$$n_0 = \frac{n}{1-r} \implies n_0 = \frac{379}{1-0.1} \implies n_0 = 421$$

The final sample size for the study ($n_o$) was a total of 421 school-aged children (SAC) (5–17 years) from each district assuming a minimum of 140 participants per school. Schools were selected from communities with proximity to freshwater bodies.

After selecting the schools, a simple random sampling technique was used to choose an average of 140 SACs aged between 5 and 17 years. The selection was done using the lottery method, ensuring an equal distribution of gender [22]. A total of 840 children were targeted for recruitment into the study.

## 2.3 Ethical consideration

The study protocol was reviewed and approved by the Institutional Review Board of the Council for Scientific and Industrial Research (approval number: RPN 003/CSIR-IRB/2016). Prior to data collection, permission to conduct the study was obtained from local authorities, including community chiefs, opinion leaders, and the school administration. Written informed parental consent was obtained from parents or legal guardians, and written assent was obtained from participating children aged 12–17 years, in accordance with ethical standards and to ensure community engagement.Sample Collection and Parasitological Analysis.

Participants provided 20−30 mL of fresh urine in clean containers, collected between 10:00 AM and 2:00 PM. A 10 mL aliquot of each urine sample was processed using the filtration technique [23]. The urine was filtered through a 25 mm Millipore filter membrane (Merck KGaA, Germany) with a pore size of 12 µm. Schistosome eggs trapped on the filter membrane were then examined microscopically to detect and quantify *Schistosoma* ova. The remaining microscopy-positive urine samples were centrifuged at 2500 rpm for 5 minutes; the supernatant was discarded, and the deposits stored in individual cryovials at −20°C for further analysis.

## 2.4 Molecular analysis

**2.4.1 DNA extraction.** A 200 µL aliquot of the egg sediment from each microscopy-positive sample was washed three times in 1X Tris EDTA buffer (pH 8.0). The eggs were resuspended in 1 mL of 1 X Tris EDTA Buffer (pH = 8.0) in a petri dish. Under an inverted microscope, 10 individual eggs were removed from each sample with a Pasteur pipette, and each egg was placed into a separate 1.5 mL microcentrifuge tube containing 10 µL 1X Tris EDTA Buffer (pH 8.0). This ensured that DNA extraction was performed from a single egg, thereby avoiding mixed-template amplification. DNA was extracted from each single egg using the Quick DNA Miniprep kit (Zymo Research, USA) following the manufacturer's instructions with some modifications which involved mechanical lysis (vortexing with glass beads), incubation with proteinase K, and elution of DNA as described elsewhere [24]. Briefly, the specimen was incubated overnight with 10 µL of proteinase K and 400 µL of genomic lysis buffer after mechanical lysis with glass beads. Following that, the lysate was washed twice, and DNA was eluted (50 µL) and frozen at −20°C until used.

**2.4.2 PCR amplification.** Specific primers (S1 Table) for *S. haematobium* and *S. bovis* were used for DNA amplification. Modifications to the Asmit1 universal forward primer (Forward: TTT TTG GTC ATC CTG AGG TGT AT) [25] created ShF (Forward: TGG TCA TCC TGA GGT GTA T) and SbF (forward: TGG GCA TCC TGA GGT GTA T) for accurate species identification. Each PCR reaction included a 5 µL volume of 2x OneTaq master mix (New England Biolabs Inc., UK), 0.2 µL of 10 µM primers, and 3 µL of DNA template. Positive controls of *S. mansoni*, *S. haematobium* (DNA provided by the NIAID Schistosomiasis Resource Centre through BEI Resources, NIAID, NIH), and *S. bovis* [DNA provided by SCAN [26], as well as a negative (no template DNA) control, were used alongside each PCR to ensure specificity and accuracy of amplification.

The amplification process involved an initial 3 min denaturation at 94°C; 45 cycles of 45 seconds of denaturation at 94°C, and 30 seconds of annealing at 58.6°Cand 45 seconds extension at 72°C; followed by a final extension of 5 min at 72°C.

PCR amplicons were visualized in a VILBER Smart Imaging system by running 5 µL of each PCR product in 1.5% TAE agarose gel stained with ethidium bromide. Amplicons were sized with a 100-base pair (BP) molecular ladder and assigned as *S. haematobium and S. bovis* based on amplicon size (S1 Fig and S1 Raw Image).

 

**2.4.3 Sequencing and phylogenetic analysis.** To supplement the PCR analysis, a total of 30 PCR amplicons (ten (10) each representing distinct amplification profiles for *S. haematobium*, *S. bovis*, and *S. haematobium*/ *S. bovis*) were randomly selected, purified and sequenced bidirectionally using the Sanger sequencing method. The total number of specimen used in this analysis corresponds to the unique amplification patterns observed, not to the number of individual samples per participant. This method was used to ensure representation of the different parasite strains in the analysis.

Sequencing was performed by Inqaba Biotechnical Industries (Pty) Ltd, Pretoria, South Africa. The raw forward and reverse sequences were trimmed using FinchTV (Geospiza Inc., USA), and the reverse sequences were reverse-complemented. The cleaned sequences were aligned and further refined using Sequencher v5.4.6 (Gene Codes Corporation, USA). To confirm species identity, the processed sequences were queried against the GenBank database using the Basic Local Alignment Search Tool (BLAST).

For phylogenetic analysis, a maximum likelihood (ML) tree was constructed using the Tamura-Nei model [27] with 1,000 bootstrap replications to assess branch support. The Tamura-Nei model was selected due to its ability to account for variable nucleotide substitution rates, providing a more accurate representation of evolutionary relationships. Its use therefore provides a more realistic estimation of evolutionary distances and improves the accuracy of maximum likelihood tree construction. The tree included reference COX1 sequences retrieved from GenBank, representing *Schistosoma haematobium* (NC008074, DQ157222), Schistosoma bovis (AJ519521, OL840268), and *Schistosoma haematobium*/S. *bovis* hybrids (KT354662, MT159597). The inclusion of these reference sequences allowed for the characterization of genetic diversity among the studied populations.

To further contextualize the genetic relationships, an outgroup was incorporated into the analysis. The selected outgroup species included *Fasciola hepatica* (NC002546, AF216697), *Schistosoma mekongi* (NC002529, AF217449), *Schistosoma spindalis* (NC006067, DQ157223), *Schistosoma mansoni* (NC002545, AF216698), and *Schistosoma japonicum* (KB196417, KV196398). The inclusion of these species facilitated a broader phylogenetic comparison and provided a reference point for assessing divergence patterns among the studied *Schistosoma* species.

## 2.5 Data and statistical analysis

Data were entered into Microsoft Office Excel 2019 and imported into SPSS version 27 (IBM, USA) and GraphPad Prism 9.0 (GraphPad, San Diego, CA, USA) for analysis. Prevalence rates for *Schistosoma* infections were calculated with 95% confidence intervals. The intensity of infection was categorized into negative, light intensity (1–49 eggs/10 mL), and heavy intensity (≥50 eggs/10 mL) based on WHO thresholds [28]. Statistical comparisons between subgroups were performed using t-tests, ANOVA, and Chi-square tests, as appropriate. Differences in prevalence and intensity of infection between regions, age groups, and genders were analysed.

## 3. Results

### 3.1 Demographic characteristics

A total of 840 schoolchildren participated in the study, with 426 from Ga South and 414 from Birim North (Table 1). The mean age of participants was 11.2 years. The majority were male [n = 430, 51.19%], and children aged 9–12 years constituted a significant portion of the study population [n = 421, 50.12%].

### 3.2 Prevalence and intensity of *Schistosoma* spp infection by microscopy

Of the 840 samples analysed, 96 tested positive for *S. haematobium*, yielding a prevalence of 11.42% [95% CI: 9.26–13.96]. In Ga South, 80 out of 426 samples were positive, representing a prevalence of 18.78% [95% CI: 14.89–23.37]. In contrast, Birim North showed only 16 positive cases out of 414, corresponding to a prevalence of 3.86% [$CI_{95}$: 2.21–6.28] (Table 2). A significant difference in prevalence between Ga South and Birim North was noted [$x^2$ = 46.12, p < 0.001].

**Table 1. Demographic characteristics of the study participants.**

| Variable | Ga South n [%] | Birim North n [%] | Total n [%] |
|---|---|---|---|
| **Gender** | | | |
| Male | 217 [50.94] | 213 [51.44] | 430 [51.19] |
| Female | 209 [49.16] | 201 [48.56] | 410 [48.81] |
| **Age** | | | |
| 5–8 years | 134 [31.46] | 83 [20.04] | 217 [25.83] |
| 9–12 years | 176 [41.31] | 245 [59.42]0 | 421 [50.12] |
| 13–17 years | 116 [27.23] | 86 [20.54] | 202 [24.05] |
| Total | 426 [50.70] | 414 [49.30] | 840 [100] |

Values reported are frequencies [n] and percentages [%].

**Table 2. Distribution of S. haematobium infection (by microscopy) among study participants.**

| Variable | Ga South | | Birim North | |
|---|---|---|---|---|
| | Total | n [%, $CI_{95}$] | Total | n [%, $CI_{95}$] |
| **Total** | **426** | **80 [18.78, 14.89 – 23.37]** | **414** | **16 [03.86, 02.21 – 06.28]** |
| **Gender** | | | | |
| Male | 217 | 41[a] [22.58, 17.53 – 28.60] | 213 | 9[c] [4.23, 2.26 – 7.83]00 |
| Female | 209 | 39[a] [18.66, 13.97 – 24.50] | 201 | 7[c] [3.34, 1.65 – 6.74]00 |
| **Age group (years)** | | | | |
| 5–8 | 134 | 21[b] [15.67, 10.50 – 22.79] | 83 | 4[d] [4.82, 1.96 – 11.75] |
| 9–12 | 176 | 32[b] [18.18, 13.19 – 24.55] | 245 | 8[d] [3.25, 1.67 – 6.28]0 |
| 13–17 | 116 | 27[b] [23.27, 16.52 – 31.77] | 86 | 4[d] [4.65, 1.89 – 11.35] |

Values reported are the total number of participants [N], number of positives [n] with its prevalence [%] and corresponding 95% confidence interval [$CI_{95}$].

Column values with the same superscript do not differ significantly at $p < 0.05$.

Gender-wise, males showed the highest prevalence in both districts: 22.58% [95%CI: 17.53–28.60] in Ga South and 4.23% [95%CI: 2.26–7.83] in Birim North (Table 2). However, the prevalence of infection in males did not differ significantly from females in the Ga South [$x^2 = 0.99$, $p = 0.318$] and Birim North District [$x^2 = 0.22$, $p = 0.636$]. Regarding age, the highest prevalence in Ga South was among 13–17-year-olds [23.27%; 95% CI: 16.52–31.77], while in Birim North, it was among 5–9-year-olds [4.82%; 95% CI: 1.96–11.75] (Table 2).

Overall, 86.46% [n = 83] of the participants had light intensity infections and 13.54% [n = 13] had high intensity infection. In Ga South, 85.00% [n = 68] of the participants had light infection and most high-intensity infections were observed in participants aged 9–12 years [18.70%] (Fig 1A). In Birim North, 93.75% [n = 15] of the participants had light infection and 6.25% [n = 1] had light infection. The high-intensity infections occurred exclusively in the 9–12 age group [12.5%] (Fig 1B). Low-intensity infections predominantly affected males in both districts (Fig 1C and 1D). No significant differences in the intensity of infection were observed across age groups or between genders in either study area [p > 0.05].

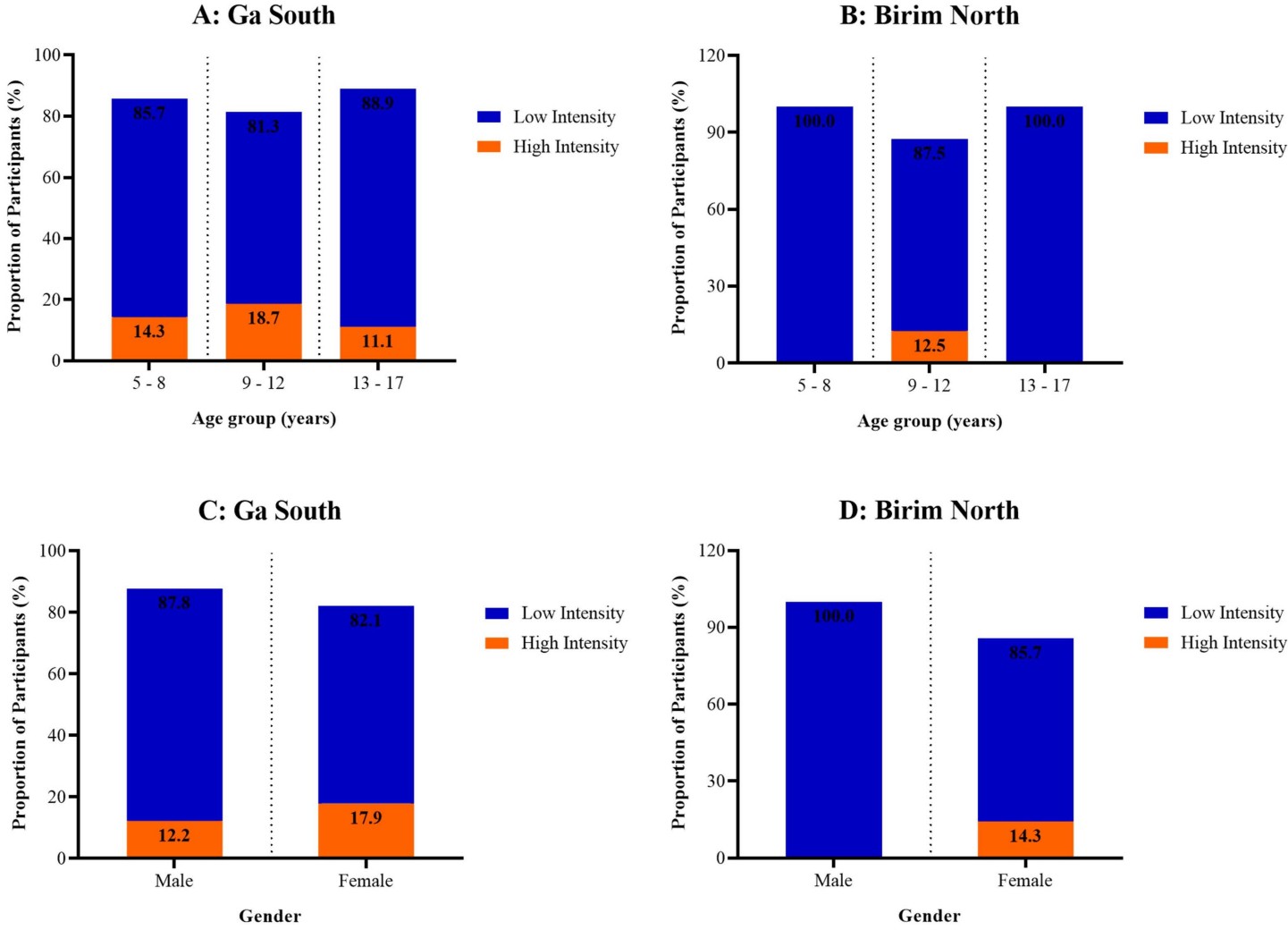

**Fig 1. Distribution of Schistosoma haematobium egg intensity among the study participants [A – B: Intensity of infection by age in Ga South and Birim North; C – D: Intensity of infection by gender in Ga South and Birim North].**

### 3.3 Species-specific prevalence by PCR

From the 13 high intensity infection identified by microscopy, 10 single eggs (*S. haematobium*-like ova) were picked individually for each participant. Thus, a total of 130 single eggs were analysed from participants with high intensity infection (refer to section 3.2). Of the 130 samples analysed 78 [60.00%, 95% CI: 51.39–68.02] of the samples were confirmed to be *S. haematobium* and 40 [30.77%, 95% CI: 23.48–39.18] were confirmed to be *S. bovis* (Fig 2). However, 12 [9.23%, 95% CI: 5.39–15.45] of the single eggs analysed were identified to have a mixed of *S. haematobium* and *S. bovis* infection (Fig 2) from PCR assays (refer to section 2.5.2). Statistically, the prevalence of *S. haematobium* detected by PCR differed significantly from that of *S. bovis* [$\chi^2 = 22.319$, $p < 0.001$] and a mixed of *S. haematobium* and *S. bovis* infection [$\chi^2 = 73.741$, $p < 0.001$] (Fig 2). Similarly, the prevalence of *S. bovis* detected was significantly different from that of the mix of *S. haematobium* and *S. bovis* (Sh/Sb) infection [$\chi^2 = 18.776$, $p < 0.001$] (Fig 2). The distribution of each parasite infra-population, the proportion of COX 1 profile of *S. haematobium*, *S. bovis* and a mix of *S. haematobium* and *S. bovis* from the 130 single eggs analysed is presented in Table 3.

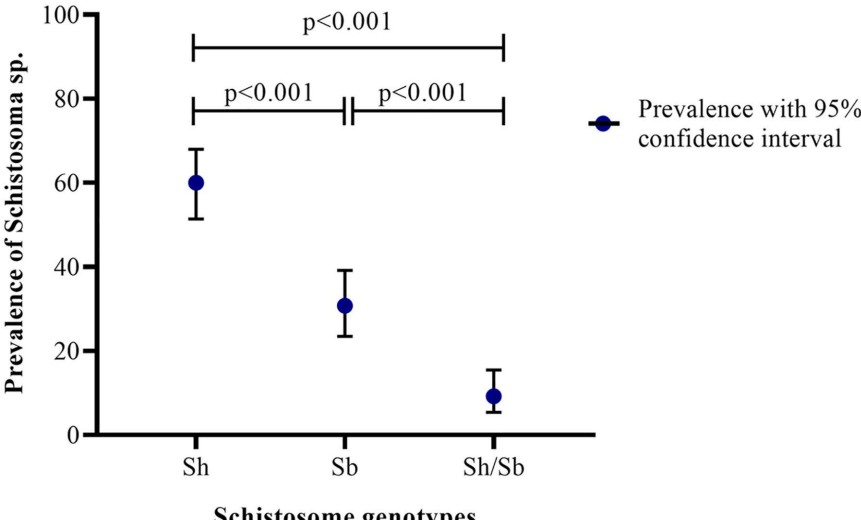

**Fig 2. Genotypes of schistosome identified in microscopy positive urine samples.** Sh: S. haematobium; Sb: S. bovis; Sh/Sb: Mixed infection.

Table 3. Characterisation of schistosomes in single egg assay using specie-specific primers.

| Participant | Total | Sh[1] | Sb[2] | Sh/Sb[3] |
|---|---|---|---|---|
| BN024 | 10 | 8 [80.00] | 1 [10.00] | 1 [10.00] |
| GS004 | 10 | 6 [60.00] | 4 [40.00] | 0 [0.00] |
| GS009 | 10 | 3 [30.00] | 7 [70.00] | 0 [0.00] |
| GS013 | 10 | 6 [60.00] | 3 [30.00] | 1 [10.00] |
| GS020 | 10 | 7 [70.00] | 1 [10.00] | 2 [20.00] |
| GS075 | 10 | 3 [30.00] | 7 [70.00] | 0 [0.00] |
| GS076 | 10 | 3 [30.00] | 5 [50.00] | 2 [20.00] |
| GS083 | 10 | 5 [50.00] | 2 [20.00] | 3 [30.00] |
| GS091 | 10 | 7 [70.00] | 2 [20.00] | 1 [10.00] |
| GS107 | 10 | 8 [80.00] | 0 [0.00] | 2 [20.00] |
| GS115 | 10 | 7 [70.00] | 3 [30.00] | 0 [0.00] |
| GS120 | 10 | 6 [60.00] | 4 [40.00] | 0 [0.00] |
| GS122 | 10 | 9 [90.00] | 1 [10.00] | 0 [0.00] |
| **Total** | **130** | **78 [60.00]** | **40 [30.77]** | **12 [9.23]** |

1.Sh represents samples with only *S. haematobium* amplification after each singleplex PCR assay of *S. haematobium* and *S. bovis*.

2.Sb represents samples with only *S. bovis* amplification afer each singleplex PCR assay of *S. haematobium* and *S. bovis*.

3.Sh/Sb represents samples that amplified for both *S. haematobium* and *S. bovis* after each singleplex PCR assay of *S. haematobium* and *S. bovis*.

## 3.4 *Schistosoma* species *Sequence* analysis

The phylogenetic analysis of *Schistosoma* isolates from urine samples revealed important insights into the genetic relationships among *S. haematobium*, *S. bovis*, and the mix of *S. haematobium* and *S. bovis* in a single isolate. The nucleotide sequences obtained in this study have been deposited in GenBank under accession numbers [PV265648-PV265673] [29].

Phylogenetic analysis based on mitochondrial cytochrome c oxidase subunit 1 (COX1) sequences reveals distinct clustering patterns among *S. haematobium*, *S. bovis*, and their hybrids in endemic communities in Ghana (Fig 3). Isolates segregate into three well-supported clades, corresponding to *S. haematobium*, *S. bovis*, and hybrid taxa. The *S. haematobium* clade includes ten isolates (GH-UR-SH01 to GH-UR-SH10), which group with reference *S. haematobium* sequences

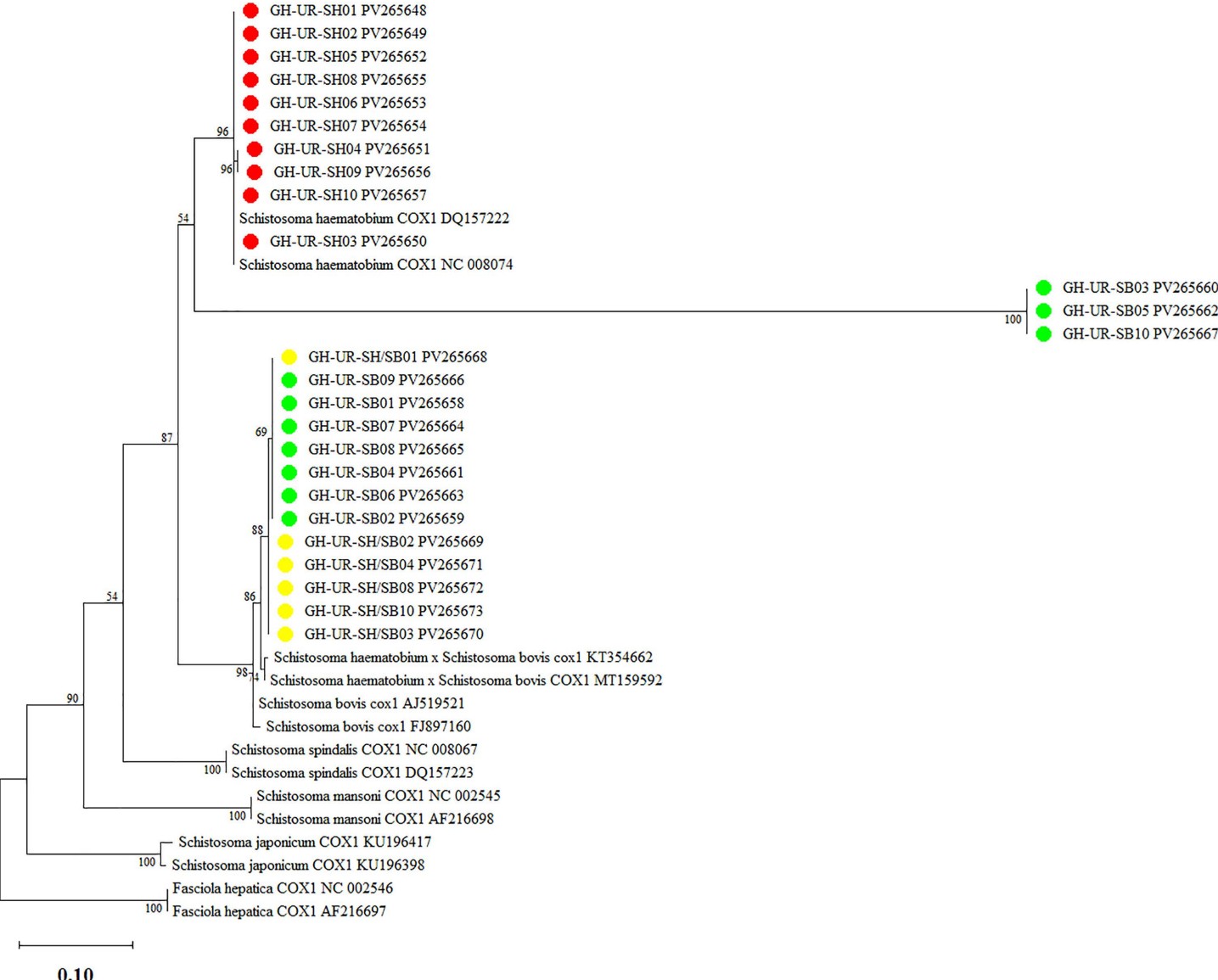

**Fig 3. Phylogenetic tree showing evolutionary relationships among *Schistosoma* species at the COX1 locus, [K: *S. haematobium* (red); F: *S. haematobium*/*S.bovis* (yellow); M: *S. bovis* (green)].** The phylogeny was inferred using the Maximum Likelihood method and Tamura-Nei (1993) model of nucleotide substitutions and the tree with the highest log likelihood (−1,667.41) is shown. The percentage of replicate trees in which the associated taxa clustered together (1,000 replicates) is shown next to the branches (Felsenstein, 1985). The initial tree for the heuristic search was selected by choosing the tree with the superior log-likelihood between a Neighbor-Joining (NJ) tree (Saitou and Nei 1987) and a Maximum Parsimony (MP) tree. The NJ tree was generated using a matrix of pairwise distances computed using the Tamura-Nei (1993) model. The MP tree had the shortest length among 10 MP tree searches, each performed with a randomly generated starting tree. The analytical procedure encompassed 40 nucleotide sequences with 316 positions in the final dataset. Evolutionary analyses were conducted in MEGA12 [30] utilizing up to 4 parallel computing threads.

(COX1 DQ157222 and COX1 NC 008074) with strong bootstrap support (96%). These isolates represent a genetically homogeneous population. A separate clade, comprising three isolates (GH-UR-SB03, GH-UR-SB05, and GH-UR-SB10), clusters with *S. bovis* reference sequences (AJ519521, FJ879160) and is supported by a bootstrap value of 100%. The presence of *S. bovis* in human-inhabited regions highlights potential zoonotic transmission risks. A third, intermediate clade comprising ten isolates (GH-UR-SH/SB01 to GH-UR-SH/SB10) formed a distinct lineage between *S. haematobium* and *S. bovis*. These sequences cluster with previously characterized *S. haematobium* × *S. bovis* hybrids (COX1 KT354662 and COX1 MT159592), with moderate to high bootstrap support (69–88%).

## 4. Discussion

This study identified significant differences in the prevalence of *S. haematobium* between the two study areas, with Ga South exhibiting a higher prevalence (18.78%) compared to Birim North (3.86%). Such disparities may be influenced by a variety of factors, including environmental conditions, water contact behaviours, and the presence of intermediate snail hosts crucial for the transmission of schistosomiasis [31]. These findings highlight the need for -targeted public health interventions that consider local ecological and socio-economic contexts.

The proportion of high-intensity infections was higher in Ga South (15.00%) compared to Birim North (6.25%), but this difference was not significant ($\chi^2 = 0.863$, $p = 0.353$). Likewise, no statistically significant differences in infection intensity were observed across age groups or between genders, although older children (13–17 years) in Ga South showed a tendency towards higher intensity infections, potentially reflecting increased exposure to contaminated water sources or water contact activities. Males exhibited higher infection rates than females in both districts; however, these differences were not statistically significant. These results suggest observable trends related to age and gender, consistent with previous studies that indicate that behavioral factors may influence exposure risk [32].

Molecular analysis confirmed that *S. haematobium* was the most prevalent species, consistent with previous literature [33]. However, the notable prevalence of *S. bovis* (33.33%) in samples initially diagnosed as *S. haematobium* by microscopy highlights significant limitations of microscopy alone in regions where multiple *Schistosoma* species co-exist [34]. This underscores the critical need for integrating molecular techniques into routine diagnostic protocols to improve accuracy in species identification. The presence of *S. bovis* in human samples also raises important concerns regarding the potential for zoonotic transmission. This finding may suggest environmental overlap between human and livestock habitats, a scenario that has been documented in other settings [35]. The molecular identification of mixed infections (11.49%) complicates the epidemiological landscape, with serious implications for disease transmission and treatment strategies. Interdisciplinary approaches to schistosomiasis management that consider both human and veterinary health are needed.

In this study, we sequenced the mitochondrial COX1 gene across *Schistosoma* species. Phylogenetic reconstruction revealed three major clades: one corresponding to *S. haematobium,* another to *S. bovis*, and a third potentially representing *S. haematobium* × *S. bovis* hybrids. These findings suggest mitochondrial introgression may occur during hybridization events. Genome recombination plays a key role in generating genotypic diversity and facilitating adaptation and speciation after interspecies hybridization [36,37]. Traditionally, this role has been considered mainly within the nuclear genome, as recombination is thought to be limited in organelles (such as mitochondria) due to their predominantly maternal inheritance [38]. Thus, heteroplasmy – the co-occurrence of both parental mtDNA in a progeny – is rare and limited, but with some exceptions allowing for rare biparental recombination to take place [39–43]. Thus, while mtDNA is generally maternally inherited in most animals, this can be affected by hybridization, leading to introgression and heteroplasmy. Further studies on this in schistosomes are warranted.

While the phylogenetic analysis were essentially to confirm the molecular findings, a limitation of this study is the use of COX1 alone in the interpretation of the phylogenetic results. Thus, the interrelatedness of the hybrids should be interpreted with caution. A more robust approach would include multilocus analyses, particularly the integration of both mitochondrial and nuclear markers, such as ITS, which would allow for better discrimination of parental lineages and hybrid

genotypes. Whole-genome sequencing could also provide a higher level of resolution in identifying introgression patterns and clarifying evolutionary relationships. Additionally, although our investigation focused on the occurrence of hybrids in humans, it is acknowledged that incorporating intermediate host snails and characterising the cercariae they transmit and their viability would provide complementary evidence of transmission dynamics. Such integrated approaches, combining both definitive and intermediate host analyses, would strengthen our understanding of the epidemiology and evolution of schistosome hybrids in endemic regions. Nonetheless, the findings from this study add to the growing evidence of genetic exchange between human and livestock schistosomes, adaptive hybrid zones and diagnostic challenges from morphologically cryptic hybrids in West Africa [44–47].

Given the implications of these findings, it is crucial to enhance surveillance efforts to monitor the prevalence of hybrids and their impact on schistosomiasis epidemiology. Understanding the ecology and transmission dynamics of both *S. haematobium* and *S. bovis* will be essential for developing comprehensive control measures. Furthermore, the identification of hybrid strains necessitates further research into their biological characteristics, including pathogenicity and treatment responses, to ensure effective management of schistosomiasis.

This study underscores the importance of integrating molecular techniques with traditional diagnostic approaches to accurately identify *Schistosoma* species and their potential hybrids. Continued research into the epidemiological implications of hybridization, along with the evaluation of current treatment strategies against a broader range of *Schistosoma* species, will be vital for addressing the complexities of schistosomiasis in endemic areas.

## 5. Conclusions

This study detected the presence of *S. bovis* in samples initially identified as *S. haematobium*, underscoring the necessity for accurate diagnostic methods to discern between species and hybrids. In regions with high *Schistosoma* prevalence, improved diagnostic approaches are vital for effective identification and treatment. Enhanced surveillance will contribute to a better understanding of schistosomiasis epidemiology, facilitating the development of more effective control strategies. Our findings provide critical insights for public health authorities in designing effective control strategies to mitigate the burden of schistosomiasis. Further research is essential to explore the implications of hybridization on disease dynamics and to refine diagnostic tools. Continued investigation into transmission pathways and the efficacy of current treatment strategies against diverse *Schistosoma* species and hybrids will be critical for effective schistosomiasis management in endemic areas.

## Supporting information

**S1 Table. Properties of COX 1 specie-specific primers used for characterization.**
(DOCX)

**S1 Fig. Post-PCR DNA gel electrophoresis analysis of three Schistosoma spp.** Lane 1–10 are loads of respective S. haematobium and S. bovis amplications as well as mix S. haematobium/S. bovis amplification in 10 single eggs from a participants' DNA sample using the primers in S1 Table. Mk represents the molecular marker; PC is the positive control, and NC is the negative control.
(TIFF)

**S1 Raw Image.**
(PDF)

## Acknowledgments

The authors wish to thank the community leaders who permitted the work to be done in the community and to all participants for participating in the study. Many thanks to the staff of the Biomedical and Public Health Unit, CSIR-Water Research Institute that provided technical help during the study.

## Author contributions

**Conceptualization:** Yvonne Ashong, Frank Twum Aboagye, Mike Yaw Osei-Atweneboana.

**Data curation:** Yvonne Ashong, Frank Twum Aboagye, Samuel Nyarko.

**Formal analysis:** Yvonne Ashong, Frank Twum Aboagye, Samuel Nyarko.

**Funding acquisition:** Mike Yaw Osei-Atweneboana.

**Investigation:** Yvonne Ashong, Frank Twum Aboagye, Samuel Nyarko.

**Methodology:** Yvonne Ashong, Frank Twum Aboagye.

**Project administration:** Samuel Armoo, Mike Yaw Osei-Atweneboana.

**Resources:** Mike Yaw Osei-Atweneboana.

**Supervision:** Alex Yaw Debrah, Mike Yaw Osei-Atweneboana.

**Validation:** Jewelna Akorli, Dziedzom K. de Souza.

**Visualization:** Jewelna Akorli, Dziedzom K. de Souza.

**Writing – original draft:** Yvonne Ashong, Frank Twum Aboagye, Isaac Owusu-Frimpong, Dziedzom K. de Souza.

**Writing – review & editing:** Yvonne Ashong, Frank Twum Aboagye, Isaac Owusu-Frimpong, Samuel Nyarko, Jewelna Akorli, Linda Batsa Debrah, Samuel Armoo, Alex Yaw Debrah, Dziedzom K. de Souza, Mike Yaw Osei-Atweneboana.

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
