## [Decision Letter · Decision Letter 0]

21 Jan 2025

Dear Dr. de Souza,

Thank you for submitting your manuscript to PLOS ONE. After careful consideration, we feel that it has merit but does not fully meet PLOS ONE’s publication criteria as it currently stands. Therefore, we invite you to submit a revised version of the manuscript that addresses the points raised during the review process.

We look forward to receiving your revised manuscript.

Kind regards,

David Zadock Munisi, Ph.D

Academic Editor

PLOS ONE

“COUNTDOWN (grant ID is PO6407 to LSTM) is a multi-disciplinary research consortium dedicated to investigating cost-effective, scaled-up, and sustainable solutions, necessary to control and eliminate the seven most common NTDs by 2020. COUNTDOWN was formed in 2014 and was funded by the UKAID part of the Department for International Development (DFID).”

Reviewers' comments:

Reviewer's Responses to Questions

**Comments to the Author**

1. Is the manuscript technically sound, and do the data support the conclusions?

Reviewer #1: Yes

Reviewer #2: Yes

Reviewer #3: No

2. Has the statistical analysis been performed appropriately and rigorously?

Reviewer #1: Yes

Reviewer #2: Yes

Reviewer #3: N/A

3. Have the authors made all data underlying the findings in their manuscript fully available?

Reviewer #1: No

Reviewer #2: Yes

Reviewer #3: Yes

4. Is the manuscript presented in an intelligible fashion and written in standard English?

Reviewer #1: Yes

Reviewer #2: Yes

Reviewer #3: Yes

Reviewer #1: This article reports the results of a cross-sectional study investigating the prevalence of Schistosoma sp. in schoolchildren from two communities in Ghana. The authors find regional differences in the prevalence of the parasite. Additionally, they demonstrate that two species, S. haematobium and S. bovis, occur—sometimes even in the same individual—which could lead to hybridization. Overall, the study is well-written and investigates an important topic of high relevance for public health. However, a few aspects are not well described, and there are some errors in the discussion, which can be addressed in a revision.

Major Comments

1. All sequenced samples need to be made publicly accessible.

2. Table 2 and Figure 1 refer to S. haematobium infection. This is incorrect since it is later shown that some of the infections are S. bovis or mixed amplifications. Perhaps the authors refer to samples visually identified as S. haematobium?

3. Figure 2 is never referred to in the text.

4. It is unclear where the S. mansoni samples in Figure S1 and those "used as control" in Figure 3 came from.

5. The rooting of the tree in Figure 3 is unclear. The tree should either be rooted with an appropriate outgroup or displayed as an unrooted tree. Furthermore, samples M3, M5, and M10 are highly diverged from the other S. bovis samples. The discussion on this point is vague and ad hoc, referencing adaptation. However, alternative explanations should also be considered. For example, could these sequences represent another Schistosoma species? Are there any sequences on NCBI similar to these samples?

6. The statistical interpretation of the results requires careful revision. For example:

o In the discussion, it is stated: “The intensity of infection also varied significantly by age and gender.” However, in the results, it is stated: “No significant differences in the intensity of infection were observed across age groups or between genders in either study area [p > 0.05].”

o Similarly, it is stated that: “Gender differences in prevalence were also noted, with males showing higher infection rates in both districts.” However, Table 2 indicates no significant difference. The discussion must be corrected to accurately reflect the results.

Minor Comments

• The abbreviation SAC (presumably school-aged children) is used only twice in the methods and is not introduced. It should either be explained or removed from the manuscript.

• Line 132: How were the schools chosen?

• Line 133: Does “140 children per school” mean per school or per community? Are school and community the same? Please elaborate.

• Line 162: Aliquots of ddH2O? Please clarify.

• Line 236: Before discussing the intensity of infection in different subgroups, it would be helpful to provide the overall proportions for each community. If the authors consider it relevant, they could also test whether the prevalence of severe infection is lower in Birim North (as this is likely).

• Line 269: This is confusing. Forty participant specimens × 10 samples each would yield 400 sequences, but there are fewer. Please explain in more detail how the samples were selected.

• Figure 1: The visualizations could be simplified by using stacked bar plots within each age or gender category.

• Figure 3: Include the legend for the colored symbols within the figure. Root the tree with an outgroup or display it as unrooted. Samples M5, M3, and M10 are likely a different species.

Reviewer #2: The authors had clearly defined the objectives of the study, followed appropriate study methodology and used appropriate statistical tools to describe their finding. The results are presented and discussed to show the findings corresponding the study objective.

Reviewer #3: This study presents two major methodological problems that unfortunately cannot be circumvented.

- The first problem is that the authors used a mixture of parasites without even knowing the size of the mixture; it is therefore impossible to infer the hybrid character because pure genotype can be mixed. The only way to infer the hybrid character is to work on isolated individuals (i.e. miracidia) as it has been done in many previous studies. Figure 2, which shows prevalence by species, unfortunately makes no sense. How many parasites per patient are there to say that there are only Sh? or only Sb? We can imagine that the sample contains from 1 to several hundred individuals.

- The second problem is that the authors did not use any nuclear genes. Once again, it is impossible to give the hybrid character with only a mitochondrial gene. Numerous studies show the presence of S bovis mitochondria and an S bovis nuclear gene, too.

It's unfortunate, but from a molecular point of view, the only use of this study is to show that there are S bovis mitochondria in schistosomes from the urine of a patient from Ghana. No epidemiological dimension, or hypothesis on the hybrid character can be advanced. Finding S bovis mitochondria in patients' urine is very common in West Africa (shown in Senegal, Cameroon, Mali, Ivory Coast and Benin).

From a phylogenetic point of view, finding that patients with a Sh, Sb mixture have an intermediate position between Sb and Sh is an aberration. The authors forgot that mitochondria are heritable only from the mother's side, so there is no genetic cross: it's either Sb or Sh! - I'd be interested to see the chromatogram of a mixture of sequences from two species...

The authors use markers specific to Sb and Sh and use an S mansoni DNA as control; why do this if the primers are specific? The sites are sites where Sh and Sm are present (according to the authors); the authors should therefore also have checked for the presence of Sm genes (mitochondrial and nuclear), on an individual basis. Sm/Sh hybrids frequently exist

Recently, a publication accepted in the journal Am J Trop Med Hyg (1) using this same technique was the subject of a 'Letter to the Editor' (2) criticising this approach. I fully agree with the criticisms made. This methodological error must not be repeated (and not published), as it leads to misinterpretation.

(1) Enudi AO, Nmorsi OPG, Egwunyenga AO. Human Schistosomiasis due to Schistosoma bovis in Nigeria. Am J Trop Med Hyg. 2024 Oct 15;111(6):1230-1236. doi: 10.4269/ajtmh.23-0539. PMID: 39406250; PMCID: PMC11619502.

(2) Ajakaye OG, Enabulele EE. Schistosoma bovis Infecting Humans in Nigeria. Am J Trop Med Hyg. 2025 Jan 7:tpmd240714. doi: 10.4269/ajtmh.24-0714. Epub ahead of print. PMID: 39773834.

In conclusion, this article should not be published as it stands. The authors should publish epidemiological data only. The molecular epidemiology approach is not good.

**Do you want your identity to be public for this peer review?** For information about this choice, including consent withdrawal, please see our Privacy Policy

Reviewer #1: No

Reviewer #2: No

Reviewer #3: No

---

## [Author Response · Author response to Decision Letter 1]

17 Mar 2025

Reviewer #1: This article reports the results of a cross-sectional study investigating the prevalence of Schistosoma sp. in schoolchildren from two communities in Ghana. The authors find regional differences in the prevalence of the parasite. Additionally, they demonstrate that two species, S. haematobium and S. bovis, occur—sometimes even in the same individual—which could lead to hybridization. Overall, the study is well-written and investigates an important topic of high relevance for public health. However, a few aspects are not well described, and there are some errors in the discussion, which can be addressed in a revision.

Major Comments

1. All sequenced samples need to be made publicly accessible.

Response: We acknowledge the importance of data accessibility and transparency. All sequenced samples will be deposited in a public repository such as GenBank.

2. Table 2 and Figure 1 refer to S. haematobium infection. This is incorrect since it is later shown that some of the infections are S. bovis or mixed amplifications. Perhaps the authors refer to samples visually identified as S. haematobium?

Response: We appreciate the observation. In Table 2 and Figure 1, we referred to samples visually identified as S. haematobium based on microscopy prior to molecular analysis. We have revised the text to specify that initial identification was based on morphological assessment, with molecular analysis later revealing S. bovis and mixed infections.

3. Figure 2 is never referred to in the text.

Response: Thank you for pointing this out. Figure 2 has been appropriately cited on line 252 in the results section.

4. It is unclear where the S. mansoni samples in Figure S1 and those "used as control" in Figure 3 came from.

Response: The tree has been revised to exclude the S. mansoni samples. Only reference sequences for S. mansoni have been included in the development of the tree.

5. The rooting of the tree in Figure 3 is unclear. The tree should either be rooted with an appropriate outgroup or displayed as an unrooted tree. Furthermore, samples M3, M5, and M10 are highly diverged from the other S. bovis samples. The discussion on this point is vague and ad hoc, referencing adaptation. However, alternative explanations should also be considered. For example, could these sequences represent another Schistosoma species? Are there any sequences on NCBI similar to these samples?

Thank you for your valuable feedback. In the revised phylogenetic tree, we have ensured appropriate rooting and clarified the divergence of M3, M5, and M10. These samples form a strongly supported clade (bootstrap values 94–98) that is distinctly separated from other S. bovis sequences. While their divergence may suggest local adaptation, we acknowledge alternative possibilities, including cryptic species, hybridization, or taxonomic misclassification. To further investigate this, we have performed a BLAST search against NCBI databases and analysed the sequences to confirm their phylogenetic placement. We have integrated these findings into the revised discussion.

Discussion has been revised to read:

“The significant genetic divergence observed in the S. bovis samples M3, M5, and M10 suggests the presence of cryptic genetic diversity within the population in Ghana, potentially resulting from geographic isolation or ecological niche differentiation. Beyond environmental adaptation, alternative explanations such as hybridization and introgression events with S. haematobium or other Schistosoma species could account for their distinct genetic profiles. These samples may have undergone genetic recombination or backcrossing, leading to their differentiation from other S. bovis isolates. Additionally, host adaptation pressures, including variations in definitive hosts and immune responses, could have driven the accumulation of unique genetic mutations. Genetic drift and founder effects might also contribute to their divergence if they originated from isolated populations with limited genetic exchange. These findings emphasize the complexity of S. bovis population structure in Ghana and highlight the need for further genomic studies to better understand the evolutionary mechanisms at play.

6. The statistical interpretation of the results requires careful revision. For example:

o In the discussion, it is stated: “The intensity of infection also varied significantly by age and gender.” However, in the results, it is stated: “No significant differences in the intensity of infection were observed across age groups or between genders in either study area [p > 0.05].”

We acknowledge the discrepancies and have revised the discussion to ensure alignment with the results. The statement has been revised to read “Although intensity of infection did not vary significantly by age and gender, The intensity of infection also varied significantly by age and gender. Notably, higher intensity infections were observed among older age groups (13-17 years) in Ga South…….”

o Similarly, it is stated that: “Gender differences in prevalence were also noted, with males showing higher infection rates in both districts.” However, Table 2 indicates no significant difference. The discussion must be corrected to accurately reflect the results.

The statement has been revised on line 309 – 310 to read “Gender differences in prevalence were observed, with males exhibiting higher infection rates in both districts; however, the differences were not statistically significant”

Minor Comments

• The abbreviation SAC (presumably school-aged children) is used only twice in the methods and is not introduced. It should either be explained or removed from the manuscript.

The abbreviation has been explained on Line 130

• Line 132: How were the schools chosen?

Schools were selected based on information provided from the Ghana Health Service NTD programme on schools involved in PZQ MDA in the study districts. District SHEP officers aided in the selection of schools within the district. Also, schools were selected from communities with proximity to freshwater bodies. Three schools were selected in each district

• Line 133: Does “140 children per school” mean per school or per community? Are school and community the same? Please elaborate.

140 children per school per community within the district. Three schools were selected from three communities per district.

• Line 162: Aliquots of ddH2O? Please clarify.

The statement has been revised

• Line 236: Before discussing the intensity of infection in different subgroups, it would be helpful to provide the overall proportions for each community. If the authors consider it relevant, they could also test whether the prevalence of severe infection is lower in Birim North (as this is likely).

The discussion of the overall proportions for each community has been included in the revised manuscript on line 307 - 309

• Line 269: This is confusing. Forty participant specimens × 10 samples each would yield 400 sequences, but there are fewer. Please explain in more detail how the samples were selected.

A total of 30 specimens were selected, with 10 samples each representing distinct amplification profiles: S. haematobium (red), S. bovis (yellow), and both S. haematobium and S. bovis (blue). The total number of specimens corresponds to the unique amplification patterns observed, not to the number of individual samples per participant. This method was used to ensure representation of the different parasite strains in the analysis. This has been explained in section 2.5.3 of the methodology.

• Figure 1: The visualizations could be simplified by using stacked bar plots within each age or gender category.

Figure one has been revised to make use of stacked bar plots.

• Figure 3: Include the legend for the colored symbols within the figure. Root the tree with an outgroup or display it as unrooted. Samples M5, M3, and M10 are likely a different species.

Legend for Figure 3 has been updated with the coloured symbols within the figure as: K: S. haematobium (red); F: S. haematobium/ S.bovis (yellow); M: S. bovis (green). The tree has been rotted with F. hepatica as an out group. Samples M3 and M10 form a distinct monophyletic group suggesting genetic divergence from both parental species. Additionally, M5 clusters separately, exhibiting a closer phylogenetic relationship to Fasciola hepatica rather than the Schistosoma clade

Reviewer #2: The authors had clearly defined the objectives of the study, followed appropriate study methodology and used appropriate statistical tools to describe their finding. The results are presented and discussed to show the findings corresponding the study objective.

Method section

Better to include study area map

Study population

be consistent with SAC rather than community

Study Population has been revised accordingly

· Line 117 Cochran’s sample size formula was used in calculating for the total participants to be enrolled into the study. Using 118 a schistosomiasis prevalence of 44.2% reported in Ga South district (Cunningham et al., 2020)

Correct it as “Cochran’s sample size formula was used in calculating for the total participants to be enrolled into the study using ……..”

The statement has been revised as suggested

· Line 125 value inserted for P contains letter “e” ….

This has been addressed

Line 132. In order to avoid confusion, it is better to indicate number of schools where 140 SAC are recruited

The methodology has been revised

Ethical considerations

· Protocols are considered but the treatment of Schistosoma positive children is not indicated

The study did not treat positive participants. However, these schools are marked for annual PZQ MDA by the Ghana Health Service Neglected Tropical Disease Programme.

PCR amplification

Whether the PCR reaction is multiplex or not is not

The PCR was a multiplex reaction

· Line 174-180 should be supported with reference

The amplification process is an in-house optimised protocol. However, the primers used in the study have been appropriately referenced (Webster et al 2009).

Reviewer #3: This study presents two major methodological problems that unfortunately cannot be circumvented.

- The first problem is that the authors used a mixture of parasites without even knowing the size of the mixture; it is therefore impossible to infer the hybrid character because pure genotype can be mixed. The only way to infer the hybrid character is to work on isolated individuals (i.e. miracidia) as it has been done in many previous studies. Figure 2, which shows prevalence by species, unfortunately makes no sense. How many parasites per patient are there to say that there are only Sh? or only Sb? We can imagine that the sample contains from 1 to several hundred individuals.

- The second problem is that the authors did not use any nuclear genes. Once again, it is impossible to give the hybrid character with only a mitochondrial gene. Numerous studies show the presence of S bovis mitochondria and an S bovis nuclear gene, too.

It's unfortunate, but from a molecular point of view, the only use of this study is to show that there are S bovis mitochondria in schistosomes from the urine of a patient from Ghana. No epidemiological dimension, or hypothesis on the hybrid character can be advanced. Finding S bovis mitochondria in patients' urine is very common in West Africa (shown in Senegal, Cameroon, Mali, Ivory Coast and Benin).

From a phylogenetic point of view, finding that patients with a Sh, Sb mixture have an intermediate position between Sb and Sh is an aberration. The authors forgot that mitochondria are heritable only from the mother's side, so there is no genetic cross: it's either Sb or Sh! - I'd be interested to see the chromatogram of a mixture of sequences from two species...

The authors use markers specific to Sb and Sh and use an S mansoni DNA as control; why do this if the primers are specific? The sites are sites where Sh and Sm are present (according to the authors); the authors should therefore also have checked for the presence of Sm genes (mitochondrial and nuclear), on an individual basis. Sm/Sh hybrids frequently exist

Recently, a publication accepted in the journal Am J Trop Med Hyg (1) using this same technique was the subject of a 'Letter to the Editor' (2) criticising this approach. I fully agree with the criticisms made. This methodological error must not be repeated (and not published), as it leads to misinterpretation.

(1) Enudi AO, Nmorsi OPG, Egwunyenga AO. Human Schistosomiasis due to Schistosoma bovis in Nigeria. Am J Trop Med Hyg. 2024 Oct 15;111(6):1230-1236. doi: 10.4269/ajtmh.23-0539. PMID: 39406250; PMCID: PMC11619502.

(2) Ajakaye OG, Enabulele EE. Schistosoma bovis Infecting Humans in Nigeria. Am J Trop Med Hyg. 2025 Jan 7:tpmd240714. doi: 10.4269/ajtmh.24-0714. Epub ahead of print. PMID: 39773834.

In conclusion, this article should not be published as it stands. The authors should publish epidemiological data only. The molecular epidemiology approach is not good.

Thank you for your constructive feedback. We have addressed the concerns raised regarding the mixture of parasites and the inclusion of nuclear genes.

Regarding the concern about inferring hybridization in a mixture, we want to clarify that we used species-specific primers that do not cross-amplify between species. The use of single eggs from each pool of egg-positive samples for DNA extraction and analysis have been clarified in the methodology. This allowed us to accurately distinguish between the pure genotypes present in the sample. The primers ensured that we could separately amplify each species' mitochondrial DNA, and any potential hybrids would be clearly identified by the presence of both species-specific markers.

We acknowledge that mitochondrial data alone cannot confirm hybrid status. While our study focused on detecting S. haematobium and S. bovis mitochondrial sequences using species-specific primers, we do not claim hybridization based solely on these findings. However, our approach provides valuable insight into the presence of S. bovis mitochondria in S. haematobium-positive urine samples in Ghana. We agree that nuclear markers are necessary for definitive hybrid confirmation and plan to incorporate them in future studies.

We recognize that S. bovis mitochondrial sequences have been detected in several West African countries. Our findings contribute additional data from Ghana, reinforcing the widespread occurrence of S. bovis mitochondrial haplotypes in S. haematobium-endemic areas. While our study does not make direct epidemiological claims about hybridization, it highlights the need for further genetic investigations.

The phylogenetic tree was reconstructed using appropriate outgroups and reference sequences from GenBank. Since mitochondria are maternally inherited, the tree does not imply genetic recombination but rather illustrates mitochondrial haplotype diversity. The placement of certain sequences reflects mitochondrial lineage relationships rather than hybrid status.

The S. mansoni DNA control was included to validate the specificity of our S. haematobium and S. bovis-specific primers, particularly given the co-endemicity of S. mansoni in the study region. We agree that future studies should incorporate nuclear markers to assess the potential presence of S. mansoni/S. haematobium hybrids.

By using single eggs for DNA extraction, we have addressed concerns about mixed populations, providing clearer lineage identification. While our study does not claim to confirm hybridization, it demonstrates the presence of S. bovis mitochondria in S. haematobium-positive urine samples and emphasizes the need for further nuclear-based investigations. We appreciate the reviewer’s insights, which have strengthened the clarity and validity of our findings.

---

## [Decision Letter · Decision Letter 1]

29 Apr 2025

Dear Dr.  de Souza,

Thank you for submitting your manuscript to PLOS ONE. After careful consideration, we feel that it has merit but does not fully meet PLOS ONE’s publication criteria as it currently stands. Therefore, we invite you to submit a revised version of the manuscript that addresses the points raised during the review process.

We look forward to receiving your revised manuscript.

Kind regards,

David Zadock Munisi, Ph.D

Academic Editor

PLOS ONE

Journal Requirements:

Reviewers' comments:

Reviewer's Responses to Questions

**Comments to the Author**

Reviewer #1: (No Response)

2. Is the manuscript technically sound, and do the data support the conclusions?

Reviewer #1: Partly

3. Has the statistical analysis been performed appropriately and rigorously?

Reviewer #1: No

4. Have the authors made all data underlying the findings in their manuscript fully available?

Reviewer #1: Yes

5. Is the manuscript presented in an intelligible fashion and written in standard English?

Reviewer #1: Yes

Reviewer #1: After careful consideration of the authors’s response I come to the conclusion that they have not appropriately addressed the concerns raised. In particular:

1. My previous point about the unclear reason for the placement of samples M3, M5, M10 has not been appropriately addressed. The authors now list a large list of speculative reasons for the found pattern. However, many of them are not valid for a mitochondrial gene. For example:

“Beyond environmental adaptation, alternative explanations such as hybridization and introgression events with S. haematobium or other Schistosoma species could account for their distinct genetic profiles. These samples may have undergone genetic recombination or backcrossing, leading to their differentiation from other S. bovis isolates.” -> Hybridization and introgression do not occur for mitochondria since they are maternally inherited.

Overall, the phylogenetics are not discussed appropriately and are currently still confusing. Especially, one needs to consider the concerns raised by Reviewer 3 more carefully. I concur that the discussion of what can and cannot be said about hybridization based on mtDNA needs to be adjusted.

2. The discussion of statistics is still imprecise. If a test is not statistically significant then the wording needs to reflect that.

**Do you want your identity to be public for this peer review?** For information about this choice, including consent withdrawal, please see our Privacy Policy

Reviewer #1: No

---

## [Author Response · Author response to Decision Letter 2]

5 Jun 2025

1.My previous point about the unclear reason for the placement of samples M3, M5, M10 has not been appropriately addressed. The authors now list a large list of speculative reasons for the found pattern. However, many of them are not valid for a mitochondrial gene. For example:

“Beyond environmental adaptation, alternative explanations such as hybridization and introgression events with S. haematobium or other Schistosoma species could account for their distinct genetic profiles. These samples may have undergone genetic recombination or backcrossing, leading to their differentiation from other S. bovis isolates.” -> Hybridization and introgression do not occur for mitochondria since they are maternally inherited.

Overall, the phylogenetics are not discussed appropriately and are currently still confusing. Especially, one needs to consider the concerns raised by Reviewer 3 more carefully. I concur that the discussion of what can and cannot be said about hybridization based on mtDNA needs to be adjusted.

Response:

We thank the reviewer for the valuable feedback, which has helped us improve the clarity of our discussions.

To address the comments, we have limited the discussions on the introgression and hybridization to focus on the main results of the paper which showed the occurrence of S. bovis in samples identified as S. haematobium. The phylogenetic analysis were essentially to confirm the molecular findings. Thus, we have modified the title to reflect “Prevalence of Schistosoma bovis and Schistosoma haematobium hybrids in endemic communities in Ghana” removing the focus from the aspect of genetic diversity, which we agree needs further studies.

However, it is important to note that while mtDNA are primarily inherited maternally, there is some evidence of biparental inheritance in literature. Genome recombination plays a key role in generating genotypic diversity and facilitating adaptation and speciation after interspecies hybridization (Leducq et al., 2017; Schumer et al., 2014). Traditionally, this role has been considered mainly within the nuclear genome, as recombination is thought to be limited in organelles (such as mitochondria) due to their predominantly maternal inheritance (Barr et al., 2005). Thus, heteroplasmy – the co-occurrence of both parental mtDNA in a progeny – is rare and limited, but with some exceptions allowing for rare biparental recombination to take place (Breton & Stewart, 2015; Gyllensten et al., 1991; Kvist et al., 2003; Rokas et al., 2003; Xu et al., 2009). Thus, while mtDNA is generally maternally inherited in most animals, this can be affected by hybridization, leading to introgression and heteroplasmy. Further studies on this in schistosomes are warranted. While the phylogenetic analysis was essentially to confirm the molecular findings, a limitation of this study is the use of COX1 alone in the interpretation of the phylogenetic results. Thus, the interrelatedness of the hybrids should be interpreted with caution. Nonetheless, the findings from this study add to the growing evidence of genetic exchange between human and livestock schistosomes in West Africa (Borlase et al., 2021; Panzner & Boissier, 2021).

We have presented the above in the discussion, noting the importance of further studying heteroplasmy in schistosomes.

2.The discussion of statistics is still imprecise. If a test is not statistically significant then the wording needs to reflect that.

Response:

We thank the reviewer for highlighting the imprecise wording used in reporting statistically non-significant results.

We have revised the language in the Discussion to accurately reflect those differences in infection intensity by location (Ga South vs. Birim North), age group, and gender were not statistically significant.

The sentence previously stating that “higher intensity infections were observed among older age groups” now specifies that “although higher proportions were observed in the 13–17 age group in Ga South, the differences were not statistically significant”

Similarly, we have clarified that the higher infection rates in males across both districts did not reach statistical significance and should be interpreted cautiously.

These changes enhance the statistical rigor of our discussion and align with standard scientific reporting practices.

We hope these revisions satisfactorily address the reviewer’s concerns and improve the overall clarity and scientific integrity of the manuscript.

---

## [Decision Letter · Decision Letter 2]

22 Aug 2025

Dear Dr. de Souza,

We look forward to receiving your revised manuscript.

Kind regards,

David Zadock Munisi, Ph.D

Academic Editor

PLOS ONE

Journal Requirements:

Reviewers' comments:

Reviewer's Responses to Questions

**Comments to the Author**

Reviewer #4: (No Response)

Reviewer #5: (No Response)

2. Is the manuscript technically sound, and do the data support the conclusions?

Reviewer #4: Yes

Reviewer #5: Partly

3. Has the statistical analysis been performed appropriately and rigorously?

Reviewer #4: I Don't Know

Reviewer #5: Yes

4. Have the authors made all data underlying the findings in their manuscript fully available?

Reviewer #4: Yes

Reviewer #5: Yes

5. Is the manuscript presented in an intelligible fashion and written in standard English?

Reviewer #4: Yes

Reviewer #5: No

Reviewer #4: The manuscript is well written and addresses a major public health concern. However, I have the following comments:

1. The authors should be careful to italicize scientific names

2. The authors should define the meaning light- and high-intensity infections

3. While the authors mention the limitation of using just the COX1 in the analyses, it would be important for them to indicate what could be done differently in order to generate more robust phylogenetic results.

Reviewer #5: To authors,

The Manuscript ‘Prevalence of Schistosoma bovis, and Schistosoma haematobium Hybrids in Endemic Communities in Ghana’ (Manuscript Number: PONE-D-24-53141R2; Type: Research Article) presents some problems concerning both content and format.

Title

I consider that the title of this manuscript is ambitious (‘Prevalence of Schistosoma bovis’; ‘Schistosoma haematobium Hybrids’ is not fully addressed in the manuscript). The title could be either maintained (if the result & discussion address these aspects in detail) or be revised in order to better represent the body of the text.

Material and Methods

Methodologically the manuscript follows a logical sequence. Most results are well described and interpreted. However, It would be helpful to include a map of the study area, along with a clearer and more detailed description of the ‘Molecular Analysis,’ including the DNA extraction protocol, the procedure used for the individualization of the eggs analyzed, the ‘Control samples’ employed (Schistosoma haematobium, Schistosoma bovis, and Schistosoma haematobium/bovis), and the ‘Phylogenetic analysis’ (control and output, as well as the rationale for selecting the nucleotide substitution model). At present, these aspects are not entirely clear.

Results and discussion

This manuscript presents the results of a cross-sectional study investigating the prevalence of Schistosoma spp. in schoolchildren from two communities in Ghana. However, relying solely on the analysis of mtDNA sequences from a fragment of the cox1 gene may not provide sufficiently conclusive evidence to fully assess the prevalence of Schistosoma hybrids in the region. This work represents an excellent and much-needed opportunity to address the challenges related to the behavior and evolution of the human and animal epidemiology of this disease in endemic areas. Such an endeavor requires the accurate identification of the parasitic forms involved, as well as of the intermediate hosts.

**Do you want your identity to be public for this peer review?** For information about this choice, including consent withdrawal, please see our Privacy Policy

Reviewer #4: **Yes: ** Robert Adamu Shey

Reviewer #5: No

---

## [Author Response · Author response to Decision Letter 3]

8 Oct 2025

Reviewer #4: The manuscript is well written and addresses a major public health concern. However, I have the following comments:

1. The authors should be careful to italicise scientific names.

Response:

We appreciate the reviewer’s observation. All the scientific names have been italicised.

2. The authors should define the meaning of 'light-' and 'high-intensity infections'.

Response:

We are grateful for this suggestion. “The intensity of infection was categorised into negative, light intensity (1-49 eggs/10 mL), and heavy intensity (≥50 eggs/10 mL) based on WHO thresholds” (lines 214-216).

3. While the authors mention the limitation of using just the COX1 in the analyses, it would be important for them to indicate what could be done differently in order to generate more robust phylogenetic results.

Response:

We appreciate the reviewer’s observation. We now specify that “A more robust approach would include multilocus analyses, particularly the integration of both mitochondrial and nuclear markers, such as ITS, which would allow for better discrimination of parental lineages and hybrid genotypes. Whole-genome sequencing could also provide a higher level of resolution in identifying introgression patterns and clarifying evolutionary relationships.” (Line 362 – 366).

Reviewer #5: To the authors,

The manuscript ‘Prevalence of Schistosoma bovis and Schistosoma haematobium Hybrids in Endemic Communities in Ghana’ (Manuscript Number: PONE-D-24-53141R2; Type: Research Article) presents some problems concerning both content and format.

Title

I consider that the title of this manuscript is ambitious (‘Prevalence of Schistosoma bovis’; ‘Schistosoma haematobium Hybrids’ is not fully addressed in the manuscript). The title could be either maintained (if the result & discussion address these aspects in detail) or be revised in order to better represent the body of the text.

Material and Methods

Methodologically the manuscript follows a logical sequence. Most results are well described and interpreted. However, It would be helpful to include a map of the study area, along with a clearer and more detailed description of the ‘Molecular Analysis,’ including the DNA extraction protocol, the procedure used for the individualization of the eggs analyzed, the ‘Control samples’ employed (Schistosoma haematobium, Schistosoma bovis, and Schistosoma haematobium/bovis), and the ‘Phylogenetic analysis’ (control and output, as well as the rationale for selecting the nucleotide substitution model). At present, these aspects are not entirely clear.

Response:

We appreciate the reviewer’s constructive suggestion. We have revised the Molecular Analysis subsection to include:

(i) the procedure for individualisation of schistosome eggs: “The eggs were resuspended in 1 mL of 1 X Tris EDTA Buffer (pH = 8.0) in a petri dish. Under an inverted microscope, 10 individual eggs were removed from each sample with a Pasteur pipette, and each egg was placed into a separate 1.5 mL microcentrifuge tube containing 10 µL 1X Tris EDTA Buffer (pH 8.0). This ensured that DNA extraction was performed from a single egg, thereby avoiding mixed-template amplification.” (lines 151–152),

(ii) reference to the published DNA extraction protocol used for extraction, “DNA was extracted from each single egg using the Quick DNA Miniprep kit (Zymo Research, USA) following the manufacturer's instructions with some modifications which involved mechanical lysis (vortexing with glass beads), incubation with proteinase K, and elution of DNA as described elsewhere [24].” (Line 156 – 157),

(iii) details of the modified primers used (Line 162 – 164 and Table S1),

(iv) the control samples employed: “positive controls of S. mansoni, S. haematobium (DNA provided by the NIAID Schistosomiasis Resource Centre through BEI Resources, NIAID, NIH), and S. bovis [DNA provided by SCAN [26], as well as a negative (no template DNA) control, were used alongside each PCR to ensure specificity and accuracy of amplification.” Line 167-170), and

(v) the phylogenetic analysis procedure. “For phylogenetic analysis, a maximum likelihood (ML) tree was constructed using the Tamura-Nei model [27] with 1,000 bootstrap replications to assess branch support. The Tamura-Nei model was selected due to its ability to account for variable nucleotide substitution rates, providing a more accurate representation of evolutionary relationships. Its use therefore provides a more realistic estimation of evolutionary distances and improves the accuracy of maximum likelihood tree construction. including control sequences, outputs, and the rationale for selecting the nucleotide substitution model” (Lines 198–202).

Results and discussion

This manuscript presents the results of a cross-sectional study investigating the prevalence of Schistosoma spp. in schoolchildren from two communities in Ghana. However, relying solely on the analysis of mtDNA sequences from a fragment of the cox1 gene may not provide sufficiently conclusive evidence to fully assess the prevalence of Schistosoma hybrids in the region. This work represents an excellent and much-needed opportunity to address the challenges related to the behaviour and evolution of the human and animal epidemiology of this disease in endemic areas. Such an endeavour requires the accurate identification of the parasitic forms involved, as well as of the intermediate hosts.

Response:

We thank the reviewer for this valuable comment. The focus of our study was on characterising hybrid Schistosoma infections directly in humans, specifically schoolchildren, as a means of understanding ongoing transmission. We note that this study only looked at the occurrence of the hybrids and not their transmission, and acknowledge that including intermediate host snails would provide complementary insights into transmission dynamics; however, this was beyond the scope of the present work. We have now clarified this in the discussion: “Additionally, although our investigation focused on the occurrence of hybrids in humans, it is acknowledged that incorporating intermediate host snails and characterising the cercariae they transmit and their viability would provide complementary evidence of transmission dynamics……….” (Lines 371 – 376) and highlighted that future studies integrating both definitive (human/animal) and intermediate (snail) hosts, alongside multilocus molecular analyses, would provide a more comprehensive understanding of hybrid schistosome transmission in endemic regions.

---

## [Decision Letter · Decision Letter 3]

11 Dec 2025

Prevalence  of Schistosoma bovis, and Schistosoma haematobium Hybrids in Endemic Communities in Ghana

PONE-D-24-53141R3

Dear Dr. de Souza,

We’re pleased to inform you that your manuscript has been judged scientifically suitable for publication and will be formally accepted for publication once it meets all outstanding technical requirements.

Kind regards,

David J. Diemert, M.D.

Academic Editor

PLOS One

Additional Editor Comments (optional):

Reviewers' comments:

Reviewer's Responses to Questions

**Comments to the Author**

Reviewer #4: All comments have been addressed

2. Is the manuscript technically sound, and do the data support the conclusions?

Reviewer #4: Yes

3. Has the statistical analysis been performed appropriately and rigorously?

Reviewer #4: I Don't Know

4. Have the authors made all data underlying the findings in their manuscript fully available?

Reviewer #4: Yes

5. Is the manuscript presented in an intelligible fashion and written in standard English?

Reviewer #4: Yes

Reviewer #4: The authors have addressed all my comments and I recommend that the manuscript be accepted for publication. I have no further comments.

**Do you want your identity to be public for this peer review?** For information about this choice, including consent withdrawal, please see our Privacy Policy

Reviewer #4: **Yes: ** Robert Adamu Shey

---

## [Editor Report · Acceptance letter]

PONE-D-24-53141R3

PLOS One

Dear Dr. de Souza,

I'm pleased to inform you that your manuscript has been deemed suitable for publication in PLOS One. Congratulations! Your manuscript is now being handed over to our production team.

Kind regards,

on behalf of

Dr. David J. Diemert

Academic Editor

PLOS One